# Sepsis-like Energy Deficit Is Not Sufficient to Induce Early Muscle Fiber Atrophy and Mitochondrial Dysfunction in a Murine Sepsis Model

**DOI:** 10.3390/biology12040529

**Published:** 2023-03-30

**Authors:** Alexandre Pierre, Claire Bourel, Raphael Favory, Benoit Brassart, Frederic Wallet, Frederic N. Daussin, Sylvain Normandin, Michael Howsam, Raphael Romien, Jeremy Lemaire, Gaelle Grolaux, Arthur Durand, Marie Frimat, Bruno Bastide, Philippe Amouyel, Eric Boulanger, Sebastien Preau, Steve Lancel

**Affiliations:** 1Univ. Lille, Inserm, CHU Lille, Institut Pasteur de Lille, U1167-RID-AGE-Facteurs de Risque et Déterminants Moléculaires des Maladies Liées au Vieillissement, F-59000 Lille, France; 2Division of Intensive Care, Hôpital Roger Salengro, CHU de Lille, F-59000 Lille, France; 3Division of Bacteriology, Biology Pathology Institute of Lille, CHU de Lille, F-59000 Lille, France; 4Univ. Lille, Univ. Artois, Univ. Littoral Côte d’Opale, ULR 7369-URePSSS-Unité de Recherche Pluridisciplinaire Sport Santé Société, F-59000 Lille, France; 5Division of Nephrology, CHU de Lille, Université de Lille, F-59000 Lille, France

**Keywords:** sepsis, skeletal muscle, mitochondria, energy deficit, pair feeding, cecal slurry injection

## Abstract

**Simple Summary:**

Sepsis is a life-threatening infection usually resulting in admission to an Intensive Care Unit, and is responsible for 1 in 5 deaths worldwide. Muscle weakness is a common and early complication of sepsis that impairs short- and long-term prognoses. It is characterized by muscle fiber atrophy and mitochondrial dysfunction. In addition, sepsis induces a negative imbalance in energy homeostasis, also associated with hospital complications. However, whether this energy deficit disrupts the metabolism in skeletal muscle has never been studied. Our study compared three groups of mice (sepsis with a spontaneous energy deficit, control without an energy deficit, and control with an energy deficit) to investigate the impact of energy debt on skeletal muscle in the first 24 h of sepsis. We demonstrated that a sepsis-like energy deficit was not in itself sufficient to alter muscle fiber size or the mitochondrial population, in contrast with sepsis. Conversely, an energy deficit led to important metabolic adaptations that were not found in sepsis.

**Abstract:**

Sepsis-induced myopathy is characterized by muscle fiber atrophy, mitochondrial dysfunction, and worsened outcomes. Whether whole-body energy deficit participates in the early alteration of skeletal muscle metabolism has never been investigated. Three groups were studied: “Sepsis” mice, fed *ad libitum* with a spontaneous decrease in caloric intake (n = 17), and “Sham” mice fed *ad libitum* (Sham fed (SF), n = 13) or subjected to pair-feeding (Sham pair fed (SPF), n = 12). Sepsis was induced by the intraperitoneal injection of cecal slurry in resuscitated C57BL6/J mice. The feeding of the SPF mice was restricted according to the food intake of the Sepsis mice. Energy balance was evaluated by indirect calorimetry over 24 h. The *tibialis anterior* cross-sectional area (TA CSA), mitochondrial function (high-resolution respirometry), and mitochondrial quality control pathways (RTqPCR and Western blot) were assessed 24 h after sepsis induction. The energy balance was positive in the SF group and negative in both the SPF and Sepsis groups. The TA CSA did not differ between the SF and SPF groups, but was reduced by 17% in the Sepsis group compared with the SPF group (*p* < 0.05). The complex-I-linked respiration in permeabilized *soleus* fibers was higher in the SPF group than the SF group (*p* < 0.05) and lower in the Sepsis group than the SPF group (*p* < 0.01). Pgc1α protein expression increased 3.9-fold in the SPF mice compared with the SF mice (*p* < 0.05) and remained unchanged in the Sepsis mice compared with the SPF mice; the *Pgc1α* mRNA expression decreased in the Sepsis compared with the SPF mice (*p* < 0.05). Thus, the sepsis-like energy deficit did not explain the early sepsis-induced muscle fiber atrophy and mitochondrial dysfunction, but led to specific metabolic adaptations not observed in sepsis.

## 1. Introduction

Sepsis is a life-threatening organ dysfunction caused by a dysregulated host response to infection [1]. It affects 49 million patients each year and is responsible for 1 in 5 deaths worldwide [2]. Its most severe form, septic shock, is associated with profound metabolic abnormalities and a mortality rate of up to 40%. Muscle weakness is a very common complication, affecting nearly all patients admitted to an Intensive Care Unit (ICU) for septic shock, and is associated with poorer short- (e.g., increased mechanical ventilation duration, length of stay, and in-hospital mortality) and long-term outcomes [3,4,5,6]. One of the main features of sepsis-induced muscle weakness is the early alteration of the mitochondrial population [7,8,9]. In 2002, Brealey et al. demonstrated that bioenergetic failure in skeletal muscle was associated with worse outcomes and that complex I activity was inversely correlated with the severity of septic shock [10]. However, sepsis-induced mitochondrial dysfunction remains poorly understood [11].

Mitochondrial quality control pathways (MtQC) maintain the mitochondrial population’s health, repairing or renewing aged or damaged mitochondria by regulating their biogenesis, dynamics, and mitophagy. Mitochondrial biogenesis is controlled by the master regulator Peroxisome proliferator-activated receptor gamma coactivator 1-alpha (PGC1α) and permits the synthesis of new mitochondria [12]. Mitochondrial dynamics, which control the transition from fused to fragmented mitochondria and vice versa, are governed by mitochondrial dynamin-related GTPases including Mitofusin-1 (Mfn1), Mitofusin-2 (Mfn2), Optic atrophy 1 (Opa1), and Dynamin-1-like protein (Drp1). While the outer mitochondrial membrane mitofusins and the inner mitochondrial membrane Opa1 are essential for fusion, Drp1 facilitates mitochondrial fission by interacting with its multiple and non-exclusive receptors on the mitochondrial outer membrane such as Mitochondrial Fission 1 protein (*Fis1*) [13,14,15]. A fine balance between these two processes is required to maintain mitochondrial homeostasis [15,16]. Mitophagy, the degradation of damaged mitochondria via autophagy, is controlled by the cytosolic Phosphatase and tensin homolog-induced kinase 1 (PINK1), which accumulates at the mitochondrial outer membrane to phosphorylate Parkin, an E3-ubiquitin ligase [17]. The lipid form of Microtubule-associated protein 1 light chain 3 B (LC3B-II—a marker of the autophagosome formation) allows the docking of the cargo receptor p62, which recognizes poly-ubiquitin chains to engulf the mitochondria [18]. BCL2/adenovirus E1B 19 kDa protein-interacting protein 3 (BNIP3), a mitochondrial outer membrane protein, also drives mitophagy by interacting directly with LC3B [19]. In case of energetic dysfunction, mitochondrial biogenesis and autophagy are activated to restore cellular ATP availability and organelle homeostasis [20,21].

In the early phase (i.e., 24–48 h, according to the European Society for Clinical Nutrition and Metabolism recommendations [22]), sepsis induces an energy deficit [23] (where energy expenditure exceeds energy intake), which is a powerful stimulus to activate proteolytic pathways in skeletal muscles [24,25]. Several observational clinical studies have shown that a negative energy balance is associated with an increased likelihood of complications and a poor prognosis in ICU [26,27,28]. Moreover, a sustained energy deficit can lead to loss of fat and muscle mass [29]. As there is an interplay between skeletal muscle homeostasis and mitochondrial population health [30,31], insufficient energy supply may impair both. However, whether a whole-body energy debt is involved in sepsis-induced mitochondrial dysfunction in skeletal muscle has never been investigated. To our knowledge, no experimental murine study has employed pair feeding to align the caloric intake of the control group with that of the sepsis group, and hence to specifically investigate the effects of energy balance on skeletal muscle, making it difficult to distinguish whether metabolic changes are induced by sepsis or by the energy restriction characteristic of the condition.

We aimed to assess whether the negative energy balance commonly observed in the early phase of sepsis is involved in mitochondrial dysfunction and muscle fiber atrophy and whether it changes the MtQC. To address this, we compared (i) a group of Sepsis mice with a control group of Sham pair-fed (SPF) mice (with both groups having identical energy intake), and (ii) these SPF mice with a control group of Sham mice fed *ad libitum* (SF) to decipher the specific effect of sepsis or its energy restriction on skeletal muscle metabolism within the first 24 h. We succeeded in reproducing the sepsis-like energy deficit in the SPF mice (assessed by indirect calorimetry). We further demonstrated that this energy debt itself was not sufficient to induce the early mitochondrial dysfunction and muscle fiber atrophy observed in the Sepsis mice. Finally, metabolic changes specifically related to energy deficit—in particular, PGC1α and autophagy signaling—were not observed in the Sepsis mice, suggesting that sepsis induced a loss of mitohormesis.

## 2. Materials and Methods

### 2.1. Murine Model of Resuscitated Sepsis

C57BL/6J male mice aged 17 to 24 weeks old (Charles Rivers Laboratories, Saint Germain Nuelles, France) were housed in a dedicated, pathogen-free animal facility (21 °C, 12 h light/dark cycles) with free access to water and food (except for the mice in the pair-feeding experiments). All procedures were approved by the local ethics committee (Animal Experimentation Ethics CEEA75, Lille, France, APAFIS#18107-2018121214148408).

Sepsis was induced by an intraperitoneal (IP) injection of cecal slurry (CS) solution, prepared at the concentration of 100 mg of stool from a batch of donor mice per 1 mL of 10% glycerol–PBS, as previously described [32,33]. The bacteriological characteristics were assessed by culture in agar dishes (bacterial load) and matrix-assisted laser desorption ionization with time-of-flight mass spectrometric detection (MALDI-TOF) (bacterial species). The same analyses were performed on blood samples from six mice collected to assess bacteremia at the 6th hour of sepsis. A preliminary set of experiments determined the cecal slurry dose required to best reproduce human sepsis. A description of the chronological criteria of the study is detailed in Appendix A. In brief, increasing doses of CS solution were injected into non-resuscitated mice to determine the minimum lethal dose. Then, this dose was tested on mice that were subsequently resuscitated (NaCl 0.9% 25 mL/kg subcutaneously, imipenem–cilastatin 40 mg/kg IP and buprenorphine 0.1 mg/kg IP every 12 h for 5 days), and adjusted to achieve a mortality rate consistent with human septic shock (n = 15) after the acute phase (Appendix A) [6,22]. The second study was designed to evaluate the impact of sepsis and caloric restriction on muscle and mitochondrial phenotypes in the early phase (Appendix A). A 10% glycerol–PBS IP injection (Sham) or CS (Sepsis) injection was performed in mice divided into three experimental groups: SF, SPF, and Sepsis. All groups were resuscitated at the 12th hour to mimic human temporality. The SF and Sepsis mice had free access to food, while the SPF mice had restricted access according to a pair-feeding procedure, in which the mice received the same food quantity eaten by the Sepsis animals (Table 1). The mice were individualized in cages to quantify food intake. Each Sepsis mouse was matched to the weight of an SPF mouse (±2 g). The food for the SPF mice was delivered at H0 and H12 the day of the experiment. A first set of mice was housed in metabolic cages with indirect calorimetry (n = 18); a second set was euthanized by cervical dislocation at the 24th hour for muscle harvesting and muscle and mitochondrial assessment (n = 24). In the latter experiment, successive batches were performed to allow high-resolution respirometry analysis within hours of sacrifice (two mice were analyzed at the same time each day).

The mice were monitored (weight, temperature, murine sepsis score (MSS)) [34] every 12 h. Temperature was measured using an infrared thermometer (Lasergrip 774, Etekcity, Anaheim, CA, USA [35]). MSS is a clinical severity score adapted to murine sepsis and assesses seven items scored from 0 to 4, where 0 is normal and 4 is the most severe grade (Appendix A). The human endpoints were an MSS score >24/28, the complete absence of movement on tactile stimulation, and a loss of >20% of initial weight. If a mouse reached these predefined endpoints, it was euthanized.

### 2.2. Metabolic Study

Body composition was measured in a non-anesthetized mouse by time-domain nuclear magnetic resonance (TD-NMR) (Minispec fl50 7.5 MHz, Bruker, Billerica, MA, USA). Total body mass was composed of free fluid mass (i.e., unbound water in the vascular and extracellular spaces), fat mass, and lean mass [36]. The lean mass included dry lean mass and lean-associated water [37]. After 72 h of acclimatization, the food intake, locomotor activity, CO_2_ production (VCO_2_), and O_2_ consumption (VO_2_) were measured using an indirect calorimetry system (Phenomaster, TSE Systems, Berlin, Germany). Food consumption was measured by weight sensors attached to the top of food containers suspended in the cage. Ambulatory activity was measured by an infrared beam system covering the horizontal surface of the cage that continuously quantified the animal’s movement. VCO_2_ and VO_2_ (mL/h) were continuously measured for 2 min per cage every 20 min and averaged over this period. The following equations were used to calculate the respiratory exchange ratio (RER) = VCO_2_/VO_2_ and the energy expenditure (kcal/h) = (3.815 + 1.232 × RER) × VO_2_) × 1000 [38]. The daily energy intake (kcal/g/day) was calculated according to the caloric value of the diet, daily energy expenditure according to energy expenditure × 24, and daily energy balance as the daily energy intake minus the daily energy expenditure; all of these measurements were normalized to body weight. A negative energy balance represents an energy deficit. The diet (U8220G10R, SAFE, France) contained 3,3 kcal/g food, 19.3% proteins, 8.4% fat, and 72.4% carbohydrate.

### 2.3. High-Resolution Respirometry on Permeabilized Muscle Fibers

The *soleus* and *extensor digitorum longus* (EDL) muscles were mechanically dissociated in cold BIOPS solution and permeabilized (50 µg/L saponin in BIOPS, gentle agitation at 4 °C for 30 and 20 min, respectively). The muscle fibers were washed twice in the mitochondrial respiration medium Mir05 and then weighed on a precision balance (CPA225D, Sartorius, Göttingen, Germany) [39]. The oxygen consumption (JO_2_) of the muscle fibers, normalized to their wet weight, was measured at 25 °C using the O2K and recorded with DatLab 7.4 software (Oroboros Instruments, Innsbruck, Austria): 2–4 mg of fibers were incubated per sealed chamber containing 2 mL of oxygenated Mir05 (O_2_ concentration close to 400 nmol/L). Mitochondrial substrates and inhibitors were sequentially injected after a steady state was reached, according to two protocols [40]. Protocol A: the addition of pyruvate (5 mM), malate (2 mM), and glutamate (10 mM) in the absence of ADP assessed the JO_2_ of the electron transport chain not coupled to ATP synthase (LEAK). The injection of ADP (5 mM) assessed the JO_2_ of the ETC coupled with ATP synthase (oxidative phosphorylation state (OXPHOS)) and preferentially driven by complex I (noted CI-CIV). The inhibition of complex I by rotenone (0.5 µM) and the addition of succinate (10 mM, a substrate of complex II) evaluated the OXPHOS JO_2_ driven by complex II (noted CII-CIV). The inhibition of complex III with antimycin A (2.5 µM) followed by the addition of ascorbate (2 mM) and tetramethyl-p-phenylenediamine dihydrochloride (Tm, 0.5 mM—respectively a Tm reducer and a CIV-specific electron donor via cytochrome c) assessed the OXPHOS JO_2_ driven by complex IV (noted CIV). Protocol B: the addition of octanoyl-L-carnitine (0.5 mM) and malate (2 mM) assessed the LEAK state, and the injection of ADP (5 mM) assessed the OXPHOS state preferentially driven by complexes I and II (noted CI + II-IV). Quality control was performed for each sample: when JO_2_ increased by +15% after cytochrome c (10 µM) addition in the OXPHOS state, the data were excluded from the analysis due to permeabilization-induced mitochondrial damage.

### 2.4. Western Blot

The *quadriceps* were homogenized in 60 µL/mg ice-cold RIPA buffer (10 mM Tris–HCl pH 7.4, 5 mM EDTA, 0.1% SDS, 150 mM NaCl, 1% sodium deoxycholate, 1% Triton) with 1 mM phenylmethanesulfonyl fluoride and a protease/phosphatase inhibitor cocktail (#5872, Cell Signaling, Danvers, MA, USA) using a Bead Mill 4 Homogenizer (Thermo Fisher Scientific, Waltham, MA, USA), and then centrifuged (15,000× *g* for 10 min at 4 °C). The supernatant protein concentration was determined with Bradford Reagent (Sigma, Burlington, VT, USA) using a microplate reader (Tristar 5, Berthold, Bad Wildbad, Germany). Forty micrograms of denatured proteins were separated in SDS polyacrylamide 12% and transferred onto polyvinylidene fluoride membranes according to the manufacturer’s instructions (Invitrogen, Waltham, MA, USA). After overnight incubation with primary antibodies and then horseradish peroxidase-conjugated secondary antibodies (see Appendix A for details), the protein complexes were revealed (Kit Clarity Western ECL substrate, Bio-Rad, Hercules, CA, USA) using chemiluminescence (Fusion X Spectra, Vilber, Marne-la-Vallée, France). The protein expression was quantified with ImageJ software (NIH, Bethesda, MD, USA, https://imagej.nih.gov/ij/) and normalized to glyceraldehyde 3-phosphate dehydrogenase (GADPH).

### 2.5. Relative Gene Expression

Frozen muscle tissue (*quadriceps*) was homogenized in TRIzol solution (Total RNA Isolation, Invitrogen, Life Technologies, Carlsbad, CA, USA) using a homogenizer and RNA extraction was performed according to the manufacturer’s instructions. The removal of DNA was performed by DNAse (DNase I, RNase-free, Thermo Fisher Scientific, Waltham, MA, USA) activated at 37 °C for 30 min and then inactivated by EDTA at 65 °C for 10 min via a thermocycler (MJ Mini Thermal Cycler, Bio-Rad, Hercules, CA, USA). One microgram of total RNA was reverse-transcribed using the High-Capacity cDNA Reverse Transcriptase Kit (Thermo Fisher Scientific, Waltham, MA, USA) in the thermocycler (25 °C for 10 min, 37 °C for 2 h, 85 °C for 5 min). the real-time detection of cDNA was performed by amplification using the PowerUp SYBR Green Master Mix kit (Thermo Fisher Scientific, Waltham, MA, USA) on a QuantStudio 3 Real-Time PCR System sequencer (Applied Biosystems, Foster City, CA, USA). The primers are detailed in Appendix A. The PCR cycles were performed according to the manufacturer’s instructions. The data were analyzed using Quant Studio Design and Analysis Software (Thermo Fisher Scientific, Waltham, MA, USA). Relative quantification was performed according to the 2^−ΔΔCt^ method.

### 2.6. Mitochondrial DNA Copy Number

Mitochondrial DNA (mt DNA)-encoded *NADH dehydrogenase 1* (*Nd1*) and *Nd2* genes—rarely subject to deletions—and nuclear DNA-encoded *Peptidyl-prolyl cis-trans isomerase* (*Ppia*)—a housekeeping gene—were amplified by qPCR using the primers detailed in Appendix A. The total DNA was isolated from the *quadriceps* with a QIAamp Fast DNA tissue kit (QIAGEN, Hilden, Germany), following the manufacturer’s instructions. The purity and concentration were measured using Nanodrop (Thermo Fisher Scientific, Waltham, MA, USA). DNA (200 pg) was used in the real-time qPCR assays, as detailed above. The relative quantification of mt DNA normalized to nuclear *Ppia* was performed according to the 2^−ΔΔCt^ method.

### 2.7. Muscle Mass and Fibers’ Cross-Sectional Area 

The wet weight of each muscle was measured using a precision balance (ME104, Mettler Toledo). The cross-sectional area (CSA) and the minimal Feret’s diameter [41] of the *tibialis anterior* (TA) muscle fibers were measured by laminin-α2 immunofluorescence labeling. Isopentane-frozen muscle, held onto a cryosection support using OCT Embedding Matrix (CellPath, UK), was transversely cross-sectioned by 10 µm using a cryostat at −20 °C (CM3050 S, Leica, Wetzlar, Germany) and then placed on slides (Superfrost Plus, Thermo Scientific, Waltham, MA, USA). Muscle sections were washed and permeabilized (0.05% PBS–Triton for 5 min, three times, at room temperature (RT)) and then blocked (3% PBS-BSA for 1 h at RT). Each section was incubated with primary and then secondary antibodies (1% PBS-BSA for 1 h at RT) (see Appendix A for details). The nuclei were counterstained with 40,6-diamidino-2-phenylindole (DAPI). The slide was mounted on a coverslip (Diamant Star, Menzel-Gläser, Braunschweig, Germany) using Vectashield anti-fading medium (Vector Labs, Berlingame, CA, USA). At least three whole muscle sections, each with at least 1000 myofibers, were imaged in mosaic for each sample by an automated slide scanner (Axioscan Z1, Zeiss, Oberkochen, Germany). After an automatic quality analysis of the myofiber images, the Feret’s diameter and the CSA were analyzed using ImageJ implemented with the MuscleJ application [42].

### 2.8. Statistical Analysis

All statistical analyses were performed with GraphPad Prism 9 (GraphPad, San Diego, CA, USA). The quantitative values are expressed as mean values ± standard error of the mean (SEM). Kaplan–Meier survival curves were compared with the log-rank test. Comparisons within a group were performed with a paired t-test. Comparisons of the SPF vs. SF groups and the Sepsis vs. SPF groups were independent, as they addressed different questions. Thus, if the continuous values followed a normal distribution (Shapiro–Wilk test), the groups were compared with one-way ANOVA and, if statistically significant, by a post hoc Fisher’s LSD test (pairwise t-test with pooled variance). Otherwise, the groups were compared with the non-parametric Kruskal–Wallis test and, if statistically significant, by a post hoc Dunn’s test. Comparisons of three groups over time were performed with a two-way ANOVA test with correction using the post hoc Tukey test for multiple comparisons. Comparisons with a *p*-value < 0.05 were considered statistically significant.

## 3. Results

### 3.1. Sepsis-Induced Changes in Body Composition Were Not Mediated by Decreased Energy Intake Alone

We first analyzed the cecal slurry solution and determined the dose to be injected to best reproduce the survival rate of human sepsis. The CS solution had a mean total bacterial burden of 5.10^5^ CFU/mL. The culture and MALDI-TOF assays identified four bacteria: *Escherichia coli*, *Enterococcus faecalis*, *Lactobacillus murinus*, and *Streptococcus haemolyticus*. The minimal lethal dose, determined using a survival experiment without resuscitation, was 400 µL of cecal slurry solution (Appendix A). This dose was thus chosen for all of the experiments of the study. Six hours after cecal slurry injection, bacteremia occurred in all mice (n = 6), of which *Escherichia coli* and *Enterococcus faecalis* were identified and considered to be the main pathogenic bacteria in our model. Finally, resuscitation every 12 h over 5 days reduced mortality to 53%, with the first death occurring 36 h after sepsis induction (Appendix A). Hence, this model was deemed suitable for studying the early phase of sepsis, and a period of 24 h after cecal slurry injection was chosen as the duration for all subsequent experiments (Appendix A). 

The mice were closely monitored to assess the human endpoints, clinical evolution, and body composition (Figure 1A). There was no difference in the physiological parameters (body weight, temperature, MSS, fat, lean, and free fluid masses) between the groups (SF vs. SPF mice or Sepsis vs. SPF mice) at baseline (Appendix A). As expected, no death occurred within 24 h. Large variations in temperature and MSS in the Sepsis group reflected the severity of the disease [34,35]. At 24 h, the temperature of the Sepsis mice dropped to 27.9 °C, while it only slightly decreased in the SPF animals compared with baseline (Figure 1B). At this time, the MSS of the Sepsis mice increased to 16. An increase of 2 points in the MSS was observed in the SPF group, resulting from decreased activity and responsiveness to stimuli (Figure 1C). Although the SPF mice experienced a drastic decrease in body weight and lean mass (by 12 and 7%, respectively) compared with baseline (*p* < 0.001), the Sepsis mice remained stable in both regards (Figure 1D,E and Appendix A; see Appendix B for further explanations). The fat mass decreased in the same range, i.e., by 30% in both the Sepsis and SPF groups (Figure 1F and Appendix A). Free fluid increased by 60% in the Sepsis group and decreased by 15% in the SPF group compared with baseline (*p* < 0.05) (Figure 1G and Appendix A). In contrast, the SF mice displayed no variation in their temperature, MSS, body weight, lean, fat, or fluid mass over the course of the experiment (Figure 1B–G). In conclusion, while decreased caloric intake among the SPF mice did not induce all of the clinical signs, nor mimic their severity in Sepsis mice over 24 h, it probably explains the loss of fat mass among the former.

### 3.2. Early Energy Deficit Was Identical between Sepsis and SPF Mice

To determine whether the pair feeding of the Sham mice reproduced the whole-body energy deficit observed during sepsis, we assessed the energy balance using indirect calorimetry (Figure 1A). The RER, energy intake, ambulatory activity, normalized oxygen consumption to lean mass (VO_2_/lean), energy expenditure, and energy balance were not different between the groups at baseline (i.e., 24 h before sepsis induction) (Figure 2A–F). The day after the intraperitoneal injection of cecal slurry or 10% glycerol–PBS, the RER was lower in the SPF group than in the SF group (0.75 vs. 0.85, *p* < 0.01), but was not different between the SPF and Sepsis groups, indicating that the sepsis-induced shift to lipid oxidation was explained by the whole-body energy deficit. The energy intake was drastically reduced in the SPF group compared with the SF group (*p* < 0.001), but was not different between the SPF and Sepsis groups (Figure 2C). Ambulatory activity, energy expenditure, and VO_2_/lean tended to decrease moderately and heterogeneously in the SPF group compared with the SF group and did not differ between the Sepsis and SPF groups (Figure 2D,E). The resulting energy balance at H24 was lower in the SPF group than in the SF group (*p* < 0.05) and negative in both the SPF and Sepsis animals, but these differences were only significant between the SF and SPF groups (Figure 2F). In conclusion, our experimental model succeeded in generating the same energy deficit in the SPF and Sepsis mice, a deficiency that was very different to the control SF animals. 

### 3.3. Sepsis-like Energy Deficit Itself Was Not Sufficient to Induce Muscle Fiber Atrophy

We then analyzed the muscle phenotype to assess whether energy deficit alone could explain the muscle fiber atrophy usually observed during sepsis. The *soleus*, EDL, TA, *gastrocnemius*, and *quadriceps* wet weights were not different between the groups (SPF vs. SF and Sepsis vs. SPF). However, the wet weight of the hind limb muscles decreased in the SPF group compared with SF group (*p* < 0.05) and was similar to the Sepsis group (Figure 3A). However, while the TA cross-sectional area and minimal Feret’s diameter did not differ between the SPF and SF groups, they were lower in the Sepsis mice compared with the SPF group (*p* < 0.05) (Figure 3B,C). In addition, the Sepsis mice had more small-area fibers (0–1000 µm^2^, *p* < 0.001) and fewer large-area fibers (3000–4000 µm^2^ and 5000–6000 µm^2^, *p* < 0.05) compared to the SPF mice and there was no difference between the SPF and SF mice (Figure 3D). The atrophy-related genes *Muscle Atrophy F-box* (*MAFbx*), *Muscle Ring-Finger Protein-1* (Mu-RF1), *Ubiquitin C* (*Ubc*), *Forkhead box protein O1* (*Foxo1*), *Forkhead box protein O3a* (*Foxo3a*), *Regulated in Development and DNA damage responses 1* (*Redd1*) and *Krüppel-like factor 15* (*Klf15*) were higher in the SPF group than the SF group and not different between the SPF and Sepsis mice. *Branched Chain Amino Acid Transaminase 2* (*Bcat2*) was not different between the groups. Therefore, within the first 24 h, the sepsis-like energy deficit led to an upregulation of atrophy-related genes in *quadriceps* and a reduced wet weight of the hind limb muscles, but was not sufficient enough to induce TA muscle fiber atrophy, unlike sepsis.

### 3.4. Sepsis-like Energy Deficit Was Not Sufficient Enough to Induce Mitochondrial Dysfunction in Skeletal Muscle

As energy balance affects mitochondrial homeostasis, we analyzed the mitochondrial respiration in the permeabilized muscle fibers. The SPF group had higher pyruvate, malate, and glutamate (PMG) complex-I-driven JO_2_ and respiratory control ratios (RCR) than the SF group in *soleus* (*p* < 0.05) (Figure 4B). Although octanoyl-carnitine-driven JO_2_ was not different between the SPF and SF groups in *soleus* and EDL (Figure 4D,E), RCR was higher in the SPF mice than the SF mice in *soleus* (*p* < 0.05) (Figure 4D). This result indicates that the sepsis-like energy deficit did not induce mitochondrial dysfunction, but rather enhanced mitochondrial respiration. In contrast, the complex-I-driven JO_2_ was lower in the Sepsis group compared with the SPF group in *soleus* and EDL (*p* < 0.01 and *p* < 0.05, respectively) and the RCR reflected these variations (*p* < 0.001 and *p* < 0.01, respectively) (Figure 4B,C). Mitochondrial function driven by fatty acid oxidation using octanoyl-carnitine was also impaired in the *soleus* fibers (*p* < 0.01 for RCR and *p* < 0.05 for CI + II-IV) (Figure 4D), though not significantly so in EDL (Figure 4E). The LEAK and other OXPHOS states (respiration driven by complex II or IV) were not modified by sepsis or caloric restriction. Overall, the sepsis-induced mitochondrial dysfunction observed in glycolytic or oxidative muscles was not explained by the whole-body negative energy balance. On the contrary, we found that this energy deficit alone increased the mitochondrial efficiency or respiration in the SPF mice, suggesting that sepsis induced by an IP injection of cecal slurry led to different adaptive mechanisms to energy debt among the Sepsis animals.

### 3.5. Sepsis-like Energy Deficit Induced Specific MtQC Adaptations Not Found in Sepsis

To understand the differences between the SPF and Sepsis animals described above, we first determined their mitochondrial biomass. The protein expression of mitochondrial respiratory chain subunits (NADH: Ubiquinone Oxidoreductase Subunit B8 (Ndufb8), Succinate Dehydrogenase Complex Iron-Sulfur Subunit B (Sdhb), Cytochrome b-c1 complex subunit 2 (Uqcr2), Mitochondrially Encoded Cytochrome C Oxidase I (Mtco1), ATP synthase F1 subunit alpha (Atp5a)) (Figure 5A) was not different between the groups (SPF vs. SF and Sepsis vs. SPF). The Voltage-Dependent Anion Channels 1/3 (VDAC1/3) (Figure 5A) did not differ between the SF group and SPF group, but were lower in the Sepsis group than in the SPF group (*p* < 0.05). In addition, the expression of *Mt-Nd1* and *Mt-Nd2* normalized to the nuclear *Ppia* were similar in the SF and SPF groups, but lower in the Sepsis group than in the SPF group (*p* < 0.05), indicating that sepsis induced mt DNA depletion (Figure 5B). Thus, the sepsis-like energy deficit is not responsible for the reduced mitochondrial biomass observed in the Sepsis mice. Since the OXPHOS protein expression remained unaltered and a patent mitochondrial function was observed in the Sepsis group, we explored whether post-translational modifications occurred. We found that proteins harboring 3-nitrotyrosine (3-NT) tripled in the Sepsis group compared with the SPF group (*p <* 0.05) (Appendix A).

Regarding mitochondrial biogenesis, *Pgc1α* mRNA expression decreased in the Sepsis group compared with the SPF group (*p* < 0.05) (Figure 5C). mRNA expressions of *Nuclear Respiratory Factor 1* (*Nrf1*), *Mitochondrial transcription factor A* (*Tfam*), and *Silent mating type information regulation 2 homolog 1* (*Sirt1*)—regulators of biogenesis—remained unchanged between groups (SPF vs. SF and Sepsis vs. SPF). The total Pgc1α protein expression increased 3.9-fold in the SPF group compared with the SF group (*p* < 0.01) and remained unchanged in the Sepsis group compared with the SFP mice (*p* = 0.08) (Figure 5A). Taken together, these results indicate that, within 24 h, sepsis starts decreasing the biogenesis signaling and mitochondrial biomass and induces an impaired Pgc1α response relative to the negative energy balance. 

Second, we explored the mitochondrial dynamics. *Mfn1*, *Mfn2*, *Opa1*, and *Drp1 mRNAs* (Figure 6A), as well as Mfn2 and Drp1 proteins (Figure 6B), remained unchanged between the groups. Nevertheless, *Fis1* mRNA increased by ~4-fold in the SPF group compared with the SF group (*p* < 0.05), and was lower in the Sepsis group than the SPF group (*p* < 0.05) (Figure 6A). Thus, the sepsis-like energy deficit mainly stimulated higher *Fis1* expression, contrary to sepsis. 

Regarding the mitophagy pathway, the mRNA expression of *Bnip3* increased 3.4-fold in the SPF group compared with the SF group (*p* < 0.05), and was lower in the Sepsis group than in the SPF group (*p* < 0.05). Although *Beclin1*, *Atg7*, and *Atg12* in the Sepsis group were all slightly higher than the SFP group (*p* < 0.05), *Unc-51-like autophagy activating kinase* (Ulk1), *Atg5*, *Atg8l*, and *Lamp2* were not different (Figure 7A). The activated form of Parkin (phospho-Ser65 Parkin) and Pink1 were not different between the groups (SPF vs. SF or Sepsis vs. SPF) (Figure 7B). LC3B-II was not different between the groups (SPF vs. SF and Sepsis vs. SPF). The relative protein ratio of LC3B-II:LC3B-I was higher in the SPF group than in the SF or Sepsis groups (*p* < 0.05) and the protein expression of p62 (a marker of the accumulation of autophagosomes not fused with lysosomes) was not different between the SPF and SF groups, but increased threefold in the Sepsis group compared with the SPF group (*p* < 0.05) (Figure 7B). Overall, different mitophagy-related responses were observed between the SPF and Sepsis groups.

## 4. Discussion

We were able to reproduce the energy imbalance observed during murine sepsis via pair-feeding of a sham group, allowing us to specifically investigate the impact of an energy deficit on the mitochondrial population and skeletal muscle homeostasis. Using a cecal slurry injection model of sepsis, we demonstrated that the sepsis-induced muscle fiber atrophy and mitochondrial dysfunction were not explained by the energy deficit typical of the early phase of sepsis. On the contrary, this sepsis-like energy deficit drove specific metabolic changes not observed in sepsis, such as enhanced mitochondrial respiration, an increase in the Pgc1α-dependent pathway, and mitophagy–autophagy signaling. 

We used a relevant experimental model of sepsis with delayed resuscitation that largely fulfills the ARRIVE guidelines and MQTiPSS recommendations for translational research in sepsis [43,44]. Our model reproduces the sepsis-induced mitochondrial dysfunction that mainly affects complex I in humans [10]. The early alteration of the mitochondrial population in skeletal muscle has been studied in endotoxemia or non-resuscitated cecal ligature puncture models of sepsis [45]. Here, we report this alteration for the first time using a medical model of infection-mediated sepsis [33]. 

As early as 1993, Kreymann et al. demonstrated that energy expenditure does not increase during the early phase of septic shock in humans, reflecting an attenuation of metabolism, in contrast to infection without organ failure where energy expenditure increases [11,46]. We were able to model this feature in our murine model of sepsis. Indeed, the energy deficit in the Sepsis mice was explained by a decrease in energy intake and an absence of an increase in energy expenditure, as observed in humans. In addition, no published study has, to our knowledge, compared Sepsis mice with SPF mice in order to analyze the effect of energy balance on skeletal muscle. In the face of reduced food intake, the SPF mice did not significantly change their energy expenditure compared with the SF mice. The energy intake and expenditure of the Sepsis mice were similar to the SPF mice. The energy balance of the SPF mice was therefore negative, in the same range as the Sepsis mice, allowing us to compare the intrinsic effect of sepsis to a sepsis-like energy deficit. Since reduced physical activity modifies energy expenditure, muscle fiber size, and mitochondrial function through MtQC alterations [30,47], we carefully recorded this confounding factor, which was not different between the Sepsis and SPF groups. Thus, physical activity was not responsible for the altered muscle metabolism. 

Although it is well known that catabolic pathways are activated in the early phase of sepsis (observable in the reduced TA CSA and the upregulation of the atrophy-related genes), body weight and lean mass did not change within the first 24 h in our sepsis model. The unchanged whole-body weight in the Sepsis mice was explained by the reduced fat mass, the increased fluid mass, and the absence of any change in lean mass. (i) The fat mass of the Sepsis mice decreased in the same range as the SPF mice, suggesting that sepsis did not alter lipolysis capacities. This is also supported by the drastic decrease in RER in both groups. (ii) The increased free fluid mass in the Sepsis mice indicated increased extracellular water and may reflect capillary leakage or kidney failure, as is commonly described in sepsis [48,49,50,51]. (iii) the lean mass measured by TD-NMR includes the water bound to macromolecules within muscles and their extracellular matrix (composed of ~75% water and ~20% protein) [36,37,52,53]. As muscle swelling and tissue edema have been well described in human sepsis [25,54,55], and since the Sepsis mice had a state of extracellular hyperhydration in our study, such abnormalities may contribute to the overestimation of muscle wet weights, lean mass, and body weight (see Appendix B for more details on TD-NMR analysis, and Appendix A, which summarizes the impact of the hydration state in our study). 

On the other hand, sepsis-like energy was responsible for an increase in the atrophy-related genes in *quadriceps*, a decrease in lean mass, the wet weight of the hind limb muscles, and body weight, but, surprisingly, not in TA fiber CSA or minimal Feret’s diameter, indicating no muscle fiber atrophy. First, since the SPF mice exhibited a state of extracellular dehydration, their lean mass and muscle wet weight could be reduced as mentioned above. Second, while the upregulation of the atrophy-related genes is required to induce muscle atrophy, it occurs before the onset of atrophy; i.e., proteins are broken down in a time-dependent manner after the transcriptional program activation [56,57,58]. It is possible that a longer period of pair feeding (e.g., the late phase of sepsis) would have led to muscle fiber atrophy. Third, PGC-1α is an important factor opposing the effects of FoxOs on muscle mass [59] and increased in the SPF mice. Fourth, the SPF and Sepsis mice both exhibited a drastic reduction in their fat mass. Fatty acid oxidation in the SPF mice may provide sufficient energy to cope with the energy deficit, in contrast with the Sepsis mice, where it resulted in a defect in their ability to consume this substrate in the skeletal muscles. Overall, the sepsis-like energy deficit may explain the upregulation of the atrophy-related genes, but was not sufficient to induce muscle fiber atrophy within the first 24 h. The mechanisms explaining the early muscle atrophy in sepsis are complex and not yet fully elucidated, and future studies should take our findings into account.

As translational research on sepsis using rodents has so far failed to result in therapeutic drugs entering clinical use over the past decades [60], further studies should consider the energy deficit generated by sepsis to improve the understanding of metabolic pathways. Pair feeding is simple to implement within an established experimental protocol using ICU animals: while the daily assessment of food intake is not time consuming, each animal must be isolated in its own cage. Nutritional treatment (e.g., with parenteral nutrition that could positively impact the energy balance) may add to the understanding of sepsis-induced mitochondrial dysfunction. However, such experiments require anesthesia and surgery, which greatly complicate the experimental procedure (i.e., central venous catheter placement) and may also induce biases, as volatile anesthetics modulate the mitochondrial population and metabolism [61,62]. Thus, pair feeding could be of real value in deciphering which molecular pathways are specifically impaired by sepsis and not merely induced by energy imbalance. Indeed, we show here that sepsis altered the mitochondrial population and MtQC pathways in skeletal muscles independently of the sepsis-like energy deficit. SPF mice countered the mitochondrial stress triggered by their energy deficit by increasing their stress resistance and enhancing their mitochondrial function, since the complex-I-linked respiration in the SPF mice was higher than in the SF mice: this is known as mitohormesis [63]. However, the Sepsis mice did not generate such cytoprotective pathways (i.e., PGC1α or BNIP3 increases), likely contributing to their mitochondrial dysfunction and loss of mitohormesis. 

Mitochondrial depletion occurs rapidly during sepsis [64,65]. In our model, the sepsis-like energy deficit was not responsible for the reduced mitochondrial biomass observed in sepsis. Indeed, the mt DNA was not different between the SF and SPF mice, whereas the mitochondrial respiration of the SPF mice was enhanced, pointing to qualitative adaptations. On the contrary, despite no change in the expression of OXPHOS subunits proteins between groups, the decrease in VDAC and the depletion of the mt DNA in the Sepsis group indicated that mitochondrial mass started decreasing, hence the sepsis-induced mitochondrial dysfunction. 

Biogenesis signaling is essential to increase mitochondrial respiration capacity and biomass under energy stress [66]. A crosstalk between PGC1α and inflammation exists in skeletal muscle: the activation of the former mitigates the latter, while inflammation-derived nuclear factor κB alters PGC1α signaling [67,68]. Using muscle biopsies from patients admitted to ICU, Carré et al. demonstrated that *PGC1*α gene expression only increased in those who will survive [69]. In our study, the gene expression of *Pgc1α* was lower in the Sepsis group than in the SPF group. As PGC1α protein expression increased ~4-fold in the muscles of the SPF mice compared with the SF mice, it was not statistically different in the Sepsis group compared with the SPF mice. Overall, these results indicated that a sepsis-like energy deficit led to the activation of biogenesis, which may maintain the mitochondrial biomass and function. Conversely, sepsis induced an insufficient biogenesis response relative to its energy deficit in skeletal muscle.

Alternatively, mitochondrial dysfunction (independent of the substrates and predominantly on complex I) may also be explained by intrinsic alterations of OXPHOS caused by nitrosative stress, which is well known in sepsis [70,71]. Brealey et al. demonstrated that muscle nitrite/nitrate concentrations correlated with the complex I activity and the severity of the disease in septic shock patients [10]. Nitrosylation also irreversibly alters complex I [72,73] and reversibly alters complex IV [74]. In accordance with these findings, we also observed higher 3-nitrotyrosine staining in the Sepsis group compared with the SPF group. 

Autophagy, a highly evolutionarily conserved process, is required to maintain muscle homeostasis [75], but its role in sepsis remains unclear [76]. Some authors have hypothesized that its activation may participate in sepsis-induced muscle wasting [77]. However, its inhibition can lead to atrophy and the accumulation of abnormal mitochondria in adult skeletal muscles [78]. Recently, using a mouse model of non-resuscitated sepsis induced by cecal ligature puncture, Leduc-Gaudet et al. demonstrated that impairing autophagy via Atg7 genetic deletion worsens both muscle atrophy and function during sepsis [79]. Our study using a model of resuscitated sepsis supports their data, as the autophagy flux of the Sepsis mice was deficient relative to the SPF mice, since LC3-II/I was lower and p62 higher, indicating the accumulation of unfused lysosomes–autophagosomes [80]. 

Mitophagy is mainly driven by the PINK1-Parkin-dependent and -independent pathways [19]. Upon mitochondrial damage, PINK1 (a sensor of mitochondrial health) phosphorylates Parkin to activate the interaction with the phagophore. In our study, while the mitochondrial populations remained dysfunctional in the Sepsis mice, PINK1 and the Phospho-Ser65 Parkin did not increase, suggesting that PINK1-Parkin-dependent mitophagy was not activated. This may be explained by nitrosylation, which impairs PINK1 function [81]. On the other hand, energy deficit is known to activate the non-canonical mitophagy pathway regulated by BNIP3 [21]. While the energy debt increased *Bnip3* 3.4-fold in the SPF mice, the Sepsis animals did not attain this level, suggesting an impairment in the PINK1-Parkin-independent mitophagy. 

Mitochondrial dynamics are also involved in the mitophagy process as mitochondrial fission, mainly controlled by Drp1 and *Fis1*, is required to remove dysfunctional mitochondria by autophagy [82]. Despite no difference in Drp1, the sepsis-like energy deficit induced an important increase in *Fis1* mRNA not observed in the Sepsis mice. High levels of *Fis1* promote mitochondrial fragmentation and trigger autophagy [83]. Taken together, our data suggest that sepsis impairs the MtQC, leading to an insufficient renewal of the mitochondrial population in skeletal muscles. However, future studies should specifically interrogate the modulation of biogenesis and mitophagy/autophagy.

Finally, the nutrition of ICU patients in the early phase of sepsis is a major issue. Whether artificial hypo- or normocaloric nutrition is the best treatment has been debated by clinicians for many years, and the discussion is ongoing [84,85,86]. The use of artificial hypocaloric nutrition in the early phase of sepsis increases the energy deficit in humans compared with normocaloric nutrition—which is currently recommended in humans [22]. Thus, clinicians may question the intrinsic effect of the energy deficit induced by restricted nutrition, particularly on metabolic tissues such as skeletal muscle. Concerns that an energy deficit may worsen muscle status should be considered in relation to our findings. Despite the limitations of a murine study with respect to the translation of results to humans, we took advantage of the metabolic response of the Sepsis mice that had a spontaneous decrease in caloric intake to shed light on the impact of a sepsis-like energy deficit. Our study provides molecular evidence that the energy debt alone cannot explain the sepsis-induced disruption of muscle within the first 24 h. 

## 5. Conclusions

In a cecal slurry injection, mouse model of sepsis with delayed resuscitation, early sepsis-induced muscle fiber atrophy and mitochondrial dysfunction were not solely attributable to the energy deficit. In light of the energy deficit and the mitochondrial dysfunction, the biogenesis and autophagy signaling pathways were insufficiently activated in sepsis. 

## Figures and Tables

**Figure 1 biology-12-00529-f001:**
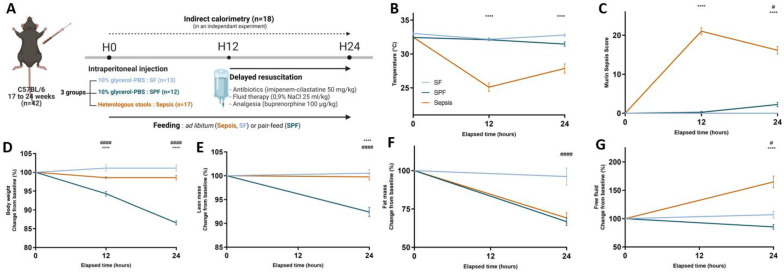
Clinical evolution after cecal slurry injection. (**A**) Timeline and experimental design of the main study: 42 mice were injected with 400 µL of cecal slurry solution or 10% PBS–glycerol solution at H0 and were resuscitated at H12. Sham-fed (SF) and Sepsis mice were fed *ad libitum* and Sham pair-fed (SPF) mice received the same amount of food as the Sepsis mice. Muscles were harvested at H24. Eighteen mice were subjected to indirect calorimetry in an independent experiment after acclimatization in specific chambers. (**B**,**C**) Temperature and murine sepsis score (MSS) were respectively lower and higher in the Sepsis group than the SPF group, but were not different between the SPF and SF groups except for the MSS at 24 h, which increased by 2 points in the SPF group (*p* < 0.05). (**D**,**E**) Change from baseline in body weight and lean mass of SPF mice significantly decreased over time compared with Sepsis and SF mice. (**F**) Change from baseline in fat mass of the Sepsis group was not different from that seen in the SPF group and both decreased in the same ranges. (**G**) Change from baseline in free fluid drastically increased in the Sepsis mice compared with the SPF mice, but was not different between SPF and SF mice. Light-blue line for the SF group (n = 13), dark-blue line for the SPF group (n = 12), and orange line for the Sepsis group (n = 17 and 13 for body composition). Data are expressed as means with SEM and analyzed using a two-way ANOVA test. # SPF vs. SF and * Sepsis vs. SPF. # *p* < 0.05, **** *p* < 0.0001, #### *p* < 0.0001.

**Figure 2 biology-12-00529-f002:**
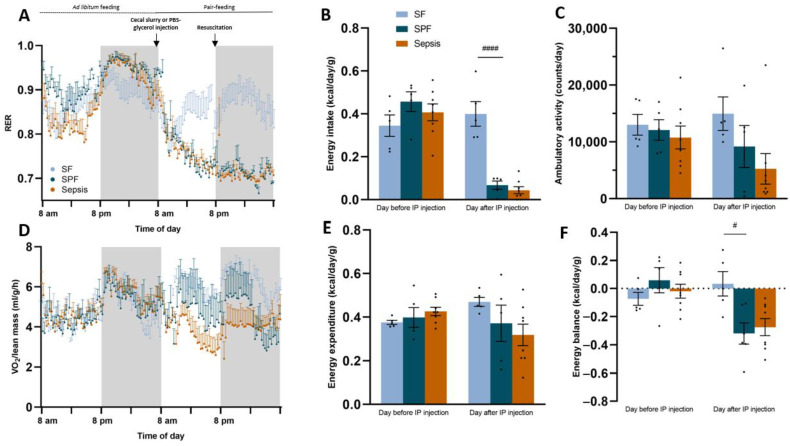
Metabolic study revealed that pair feeding induced an energy deficit identical to sepsis. The day before or after the intraperitoneal (IP) injection refers to the 24 h before or after the IP injection of PBS–glycerol or cecal slurry. (**A**) Respiratory exchange ratio (RER) remained at 0.85 in the Sham-fed (SF) group, but decreased from 0.9 to 0.75 in both the Sham pair-fed (SPF) and Sepsis groups after IP injection. (**B**) Daily energy intake was drastically reduced in SPF and Sepsis groups after IP injection (see Appendix A for the food intake expressed in grams of pellets per day). (**C**–**E**) Ambulatory activity, systemic oxygen consumption (VO_2_) normalized to lean mass, and daily energy expenditure tended to decrease in SPF compared with SF mice, but were not different between SPF and Sepsis mice after IP injection. (**F**) Daily energy balance was lower in the SPF group than the SF group and negative in the SPF and Sepsis groups, resulting in an energy deficit in the SPF and Sepsis groups. Light-blue bars for the SF group (n = 5), dark-blue bars for the SPF group (n = 5), and orange bars for the Sepsis group (n = 8). Data are expressed as mean values with SEM and compared using a Kruskal–Wallis test with post hoc Dunn’s test. # SPF vs. SF and * Sepsis vs. SPF. # *p* < 0.05, #### *p* < 0.0001.

**Figure 3 biology-12-00529-f003:**
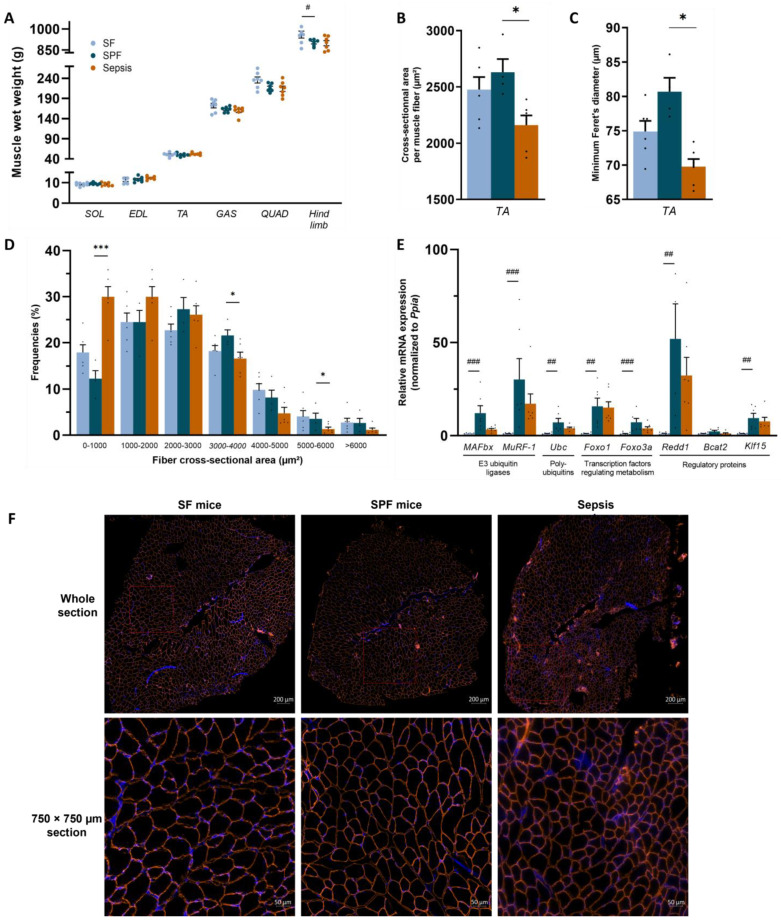
Sepsis-like energy deficit led to atrophy-related gene upregulation, but was not sufficient enough to induce muscle fiber atrophy. *SOL*, *EDL*, *TA*, *GAS*, and *QUAD* refer to *soleus*, *extensor digitorum longus*, *tibialis anterior*, *gastrocnemius*, and *quadriceps*, respectively, and the term “hind limb” refers to the sum of the SOL, EDL, TA, GAS, and QUAD of both limbs. (**A**) Wet weights of *SOL*, *EDL*, *TA*, *GAS*, and *QUAD* were not different between the groups, but the hind limb muscles decreased in the SPF group compared with the SF group and was similar to the Sepsis group. (**B**,**C**) TA cross-sectional area and minimum Feret’s diameter per muscle fiber of Sepsis mice were lower compared with SPF mice, but not different between SPF and SF mice. (**D**) Fiber area distribution of TA. (**E**) mRNA expression of the atrophy-related genes *MAFBx*, *MuRF-1*, *Ubc*, *Foxo1*, *Foxo3a*, *Redd1*, and *Klf15* were higher in the SPF group than the SF group and not different between the SPF and Sepsis groups. *Bcat2* was not different between groups. (**F**) Representative images of whole-section (up) and magnification (down) cross-sectional area immunostaining by laminin α-2 in orange and DAPI in blue of SF (left), SPF (middle), and Sepsis (right) groups. Light-blue highlights the SF group (n = 6–8), dark-blue highlights the SPF group (n = 4–7), and orange highlights the Sepsis group (n = 6–7). Data are expressed as means with SEM and analyzed using a Kruskal–Wallis test with post hoc Dunn’s test. # SPF vs. SF and * Sepsis vs. SPF. * *p* < 0.05, *** *p* < 0.001, # *p* < 0.05, ## *p* < 0.01, ### *p* < 0.001.

**Figure 4 biology-12-00529-f004:**
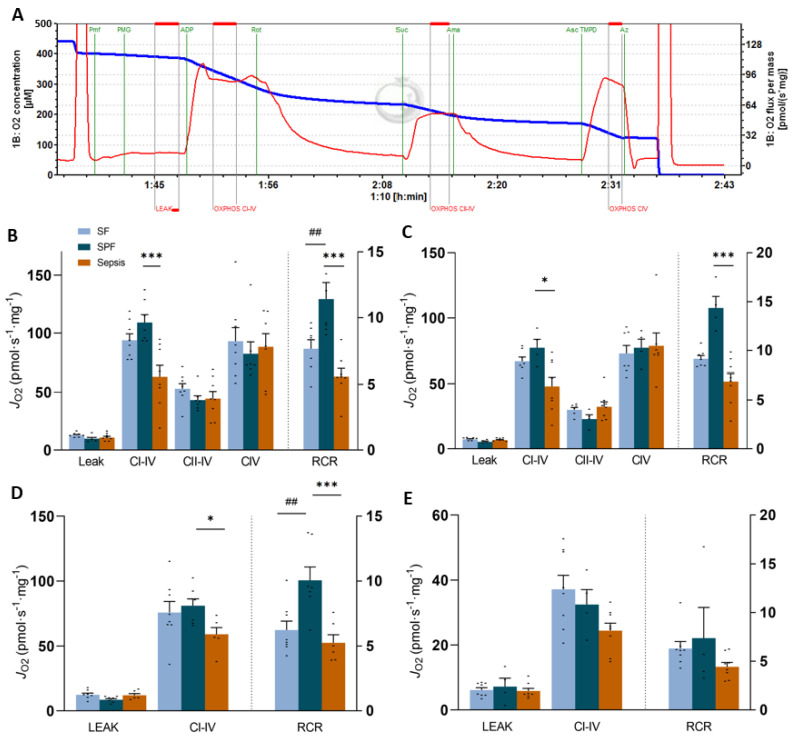
Permeabilized muscle fiber respiration in the *soleus* and EDL. Panel (**A**) represents PMG-linked JO_2_ curve after incubating permeabilized muscle fiber (pmf) from *soleus* of a Sham-fed (SF) mouse. See Appendix A for other representative curves and more details. Panels (**B**,**C**) represent *soleus* (**B**) and EDL (**C**) pmf oxygen consumption (JO_2_) recorded after sequential injections: pyruvate (5 mM), malate (2 mM), and glutamate (10 mM) (PMG) (LEAK); ADP (5 mM) (CI–IV for oxidative phosphorylation (OXPHOS) state driven by complex I); rotenone (Rot, 0.5 µM) and succinate (Suc, 10 mM) (CII–IV for OXPHOS state driven by complex II); antimycin A (Ama, 2.5 µM), ascorbate (Asc, 2 mM) and TMPD (0.5 mM) (CIV for OXPHOS state driven by CIV). Panels (**D**,**E**) represent *soleus* (**D**) and EDL (**E**) pmf JO_2_ recorded after sequential injections: octanoyl-carnitine (0.5 mM) and malate (2 mM) (LEAK), and then ADP (5 mM) (CI–IV). The respiratory control ratio (RCR) is plotted on the right *Y*-axis in each graph. One value excluded due to cytochrome c elevation in B–D. Light-blue bars for the SF group (n = 7), dark-blue bars for the SPF group (n = 4–7), and orange bars for the Sepsis group (n = 7–8). Data are expressed as mean values with SEM and analyzed by one-way ANOVA with post hoc Fisher’s LSD test (**A**,**C**) or a Kruskal–Wallis test with a post hoc Dunn’s test (**B**,**D**). # SPF vs. SF and * Sepsis vs. SPF. * *p* < 0.05, ## *p* < 0.01, *** *p* < 0.001.

**Figure 5 biology-12-00529-f005:**
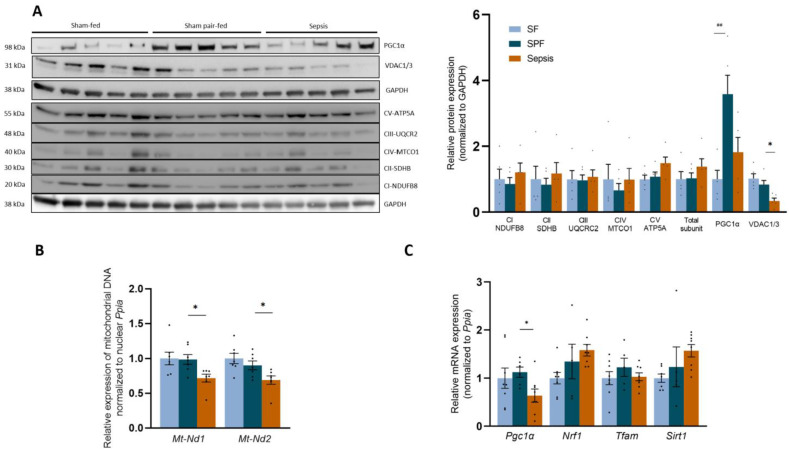
Sepsis-like energy deficit induced specific biogenesis adaptations in skeletal muscle not observed in sepsis. (**A**) Protein expression of different subunits of the respiratory chain complexes was not significantly different between groups. VDAC was lower in the Sepsis group than in the Sham pair-fed (SPF) group. Total PGC1α protein expression was higher in the SPF group than in the Sham-fed (SF) group and remained unchanged in the Sepsis group compared with the SPF group. (**B**) Mitochondrial DNA expression of *Nd1* and *Nd2* normalized to nuclear *Ppia* decreased in the Sepsis group compared with the SPF group. (**C**) mRNA expression of the biogenesis actors *Nrf1*, *Tfam*, and *Sirt1* was not different between groups, but *Pgc1α* was lower in the Sepsis group than in the SPF group. Light-blue bars for SF mice, dark-blue bars for SPF mice, and orange bars for Sepsis mice (n = 5–8 per group). Data expressed as means with SEM and compared using a Kruskal–Wallis test with post hoc Dunn’s test. # SPF vs. SF and * Sepsis vs. SPF. * *p* < 0.05, ## *p* < 0.01.

**Figure 6 biology-12-00529-f006:**
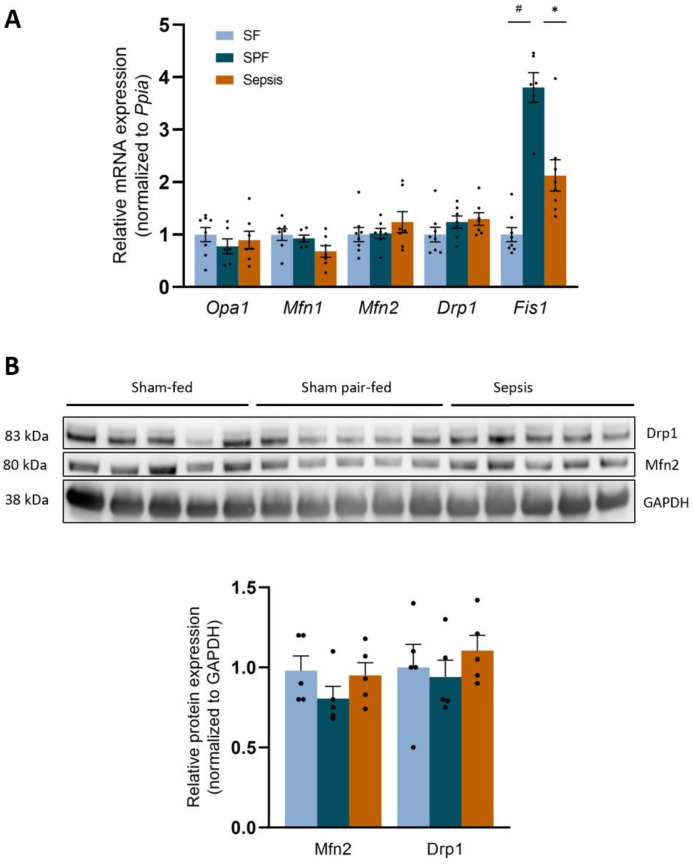
Sepsis-like energy deficit led to different mitochondrial dynamics signaling compared with Sepsis. (**A**) mRNA expression of the main mitochondrial dynamics players *Opa1*, *Mfn1*, *Mfn2*, *Drp1* were not different between groups, but *Fis1* increased in the SPF group compared with the SF group and was lower in the Sepsis group compared with the SPF group. (**B**) Protein expression of Drp1 and Mfn2 did not differ between groups. Light-blue bars for SF mice, dark-blue bars for SPF mice, and orange bars for Sepsis mice (n = 5–8 per group). Data expressed as means with SEM and compared using a Kruskal-Wallis test with post hoc Dunn’s test. # SPF vs. SF and * Sepsis vs. SPF. * *p* < 0.05, # *p* < 0.05.

**Figure 7 biology-12-00529-f007:**
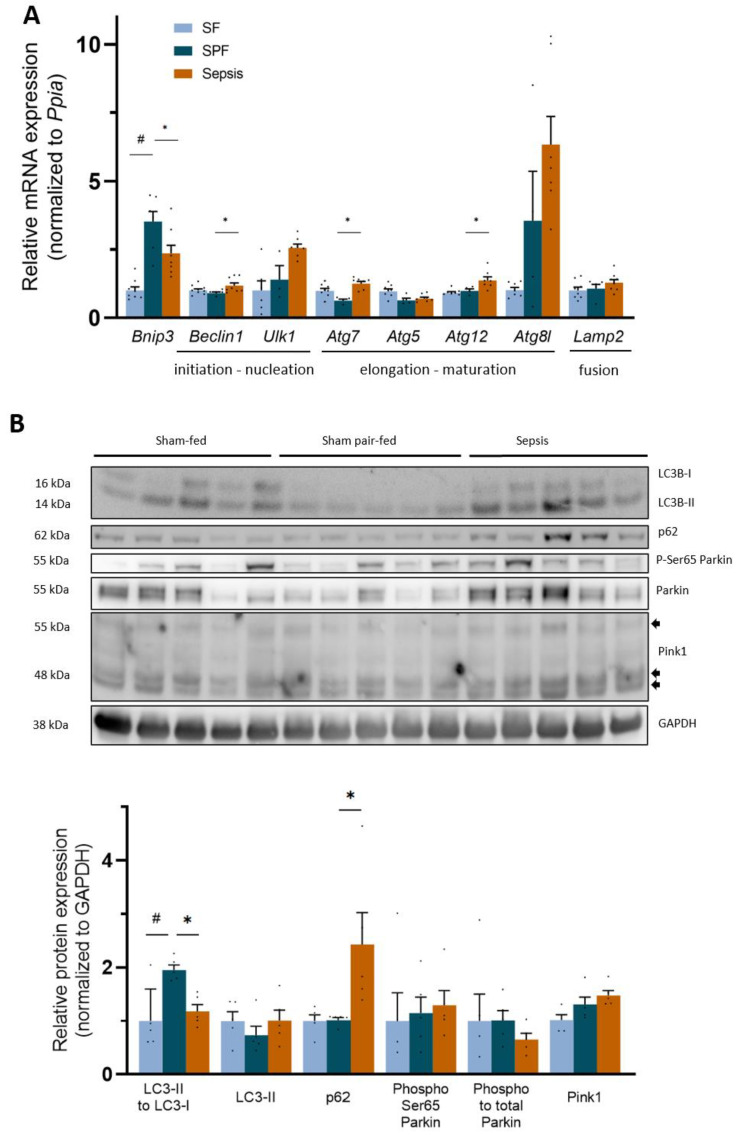
Sepsis-like energy deficit resulted in different autophagy signaling compared with sepsis. (**A**) mRNA expression of the key mito- and autophagy actor *Bnip3* increased threefold in the SPF group compared with the Sham-fed (SF) group and was lower in the Sepsis group than the SPF group. *Atg7*, *Atg12*, and *Beclin1* increased in Sepsis mice compared with SPF mice, while *Ulk1*, *Atg5*, *Atg8l*, and *Lamp2* were not different between groups. (**B**) Protein expression of the main mito- and autophagy actors: relative ratio of LC3B-II:LC3B-I decreased in Sepsis mice compared with Sham pair-fed (SPF) mice, while p62 increased. LC3B-II, Ser65 phospho-Parkin, ratio of phospho: total Parkin, Pink1, Mfn2, and Drp1 did not differ between groups. Light-blue bars for SF mice, dark-blue bars for SPF mice, and orange bars for Sepsis mice (n = 5–8 per group). Arrows indicate different forms of Pink1. Data expressed as means with SEM and compared using a Kruskal–Wallis test with post hoc Dunn’s test. # SPF vs. SF and * Sepsis vs. SPF. * *p* < 0.05, # *p* < 0.05.

**Table 1 biology-12-00529-t001:** Interventions according to the three experimental groups.

Group Designation	Group Acronym	Intraperitoneal Injection	Feeding	Aim
Sham fed	SF	10% glycerol–PBS	Free access	To control the energy balance
Sham pair-fed	SPF	10% glycerol–PBS	Restricted	To reproduce the sepsis- like energy deficit
Sepsis	Sepsis	Heterologous stools	Free access	To assess the effect of sepsis on metabolic pathways

## Data Availability

Not applicable.

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
