# Peer review of "Sepsis-like Energy Deficit Is Not Sufficient to Induce Early Muscle Fiber Atrophy and Mitochondrial Dysfunction in a Murine Sepsis Model"

_biology, 2023, doi:10.3390/biology12040529_

Round 1

Reviewer 1 Report

In my opinion this manuscript is not able to provide strong data supporting the conclusion. The paper is not well written. More importantly it opens up to a lot of questions that are not supported by novel findings (for example from authors: “Line 421 Pgc1α regulation and mitophagy may be impaired”; Line 454“the mitochondrial stress of septic mice did not generate such cytoprotective pathways (i.e., PGC1α increase) and may be responsible for the mitochondrial dysfunction, suggesting that sepsis led to the loss of the mitohormesis”; Line 395 Taken together, these data suggested that sepsis and its energy deficit did not change mitochondrial dynamics and biomass. However, although sepsis slightly upregulated (significantly?) Pgc1α upstream signaling and mito-autophagy pathways, it appeared to be insufficient relative to the observed energy deficit; Line 463: “These results suggested that sepsis led to an insufficient biogenesis response in skeletal muscle”).

Of course they can not conclude with: Line 26-29 The authors write: ”energy deficit led to important metabolic adaptations – probably to better cope with this stress, not found in sepsis. Since hypocaloric nutrition is more and more used in Intensive Care Unit patients suffering from sepsis, concerns that energy deficit may worsen muscle status has to be considered in relation to our results”.

The authors started from a clinical evaluation after cecal slurry injection.

Line 255-256 the authors write that the temperature “slightly decreased in the SPF animals, compared to baseline”. If it is not statistically significant they can not write this sentence.

Figure 1B. Present data as dots-box plot. The same should be done also for the panels D to G. Why the authors compare everything with 100% basal condition (0 hour)? Probably there is too much variability among mice at 0h. This variability could be a big problem.

Line 273-275: The authors write “Temperature and murine sepsis score (MSS) were higher in the sepsis group than the SPF group and were not different between SPF and SF groups except for the MSS at 24 hours which slightly increased in the SPF group” but the temperature in the sepsis group is LOWER.

Data in Figure 2 are not clear.

The authors write:“Data are expressed as mean values with SEM (A, D) and SD (B, C, D, E) and analyzed by post-hoc Dunn’s test”. All the data should be presented as mean values with SEM. For results in D which error is shown:SEM or SD?

Do the authors perform the Dunn’s test after a significant Kruskal-Wallis test?

More importantly analysis of covariance (ANCOVA) should be performed.

From Line 430: In the face of reduced food intake, SPF mice slightly decreased their energy expenditure similar to septic mice…… allowing us to study the intrinsic effect of sepsis or its energy deficit”. The authors based their study on a not significant data (Fig.2E). It is really difficult to understand.

In Fig3. The authors can not present the quantifications without showing representative images used for quantifications. There is not consistency in the way the data are showed. Why quantifications are performed in TA while this muscle is missing in the data of panels A and B?

More importantly the authors conclude “sepsis-induced muscle atrophy cannot be explained by the sole energy deficit, indicating different underlying mechanisms” but they didn’t show the atrophy in muscle!

In Fig.4 the authors should show the OCR traces.

What about the spare respiratory capacity?

Since Respiratory control rate changes independently of the substrate used this means that it is due to an intrinsic property of mitochondria. Please discuss this.

Fig 5. The blots have poor quality. The OXPHOS are not clearly detected.

Line 376-7  the authors write “suggesting a preserved mitochondrial biomass in skeletal muscle of septic mice”. The authors must check in details whether or not mitochondrial mass is preserved in septic mice. Western blot detecting TOMM20 and CS should be informative.

Line 379: “…tended to be lower in the sepsis group than the SFP group (p=0.08)”. The authors should increase the number of analyzed mice in order to understand if this tendency is correct or in not significant.

It is well known (for example check Mizushima and Yoshimori, Authophagy 2007; Kaludercic et al, Cardiovascular Research 2020) that the comparison of LC3-I and LC3-II for ratio determinations may not be appropriate; rather, only the amount of LC3-II can be compared between samples. Then the authors should quantify only LC3II.

Line 475-477 The authors conclude that “ … the autophagy flux of septic mice was also deficient relative to SPF mice since LC3-II/I was lower and p62 higher – a marker of the accumulation of lysosomes-unfused autophagosomes”. The levels of autophagy or mitophagy cannot be assessed simply by studying the expression levels of the involved proteins and the blots of LC-II are inconclusive in this respect. Autophagy and mitophagy can be studied in a dynamic fashion in the presence and absence of specific inhibitors.

The findings could be corroborated with additional experiments, the most important of all would consist in a more direct demonstration that altered autophagy activity, and particularly of mitophagy, is affected.In particular Autophagy, lysosomal activity and mitophagy can be studied in a dynamic fashion in the presence and absence of specific inhibitors.

More importantly:

Lines 90-92: The authors “aim at assessing if the negative energy balance commonly observed at the early phase of sepsis is involved in mitochondrial dysfunction and muscle wasting and whether it changes the MtQC”. The author should provide images of muscle sections and of mitochondria. In particular they

If the authors are not able to show mitochondrial morphology and  at least they should analyze the levels of all the mitochondrial shaping proteins: Opa1 with long and short forms, phospho and total Drp1, Mfn1 and Mfn2 and Fis1. In particular Opa1 regulate complex V assembly and cristae function and morphology, independently of its fusion-related GTPase activity (patten et al., 2014). Moreover Opa1 is linked to autophagy.

Moreover: Line 31 “Whether whole-body energy deficit participates in early skeletal muscle metabolism alteration has never been investigate”.

Please check and discuss this recent paper: doi: 10.1172/jci.insight.153944

Author Response

Response to Reviewer 1

In my opinion this manuscript is not able to provide strong data supporting the conclusion. The paper is not well written. More importantly it opens up to a lot of questions that are not supported by novel findings (for example from authors: “Line 421 Pgc1α regulation and mitophagy may be impaired”; Line 454“the mitochondrial stress of septic mice did not generate such cytoprotective pathways (i.e., PGC1α increase) and may be responsible for the mitochondrial dysfunction, suggesting that sepsis led to the loss of the mitohormesis”; Line 395 Taken together, these data suggested that sepsis and its energy deficit did not change mitochondrial dynamics and biomass. However, although sepsis slightly upregulated (significantly?) Pgc1α upstream signaling and mito-autophagy pathways, it appeared to be insufficient relative to the observed energy deficit; Line 463: “These results suggested that sepsis led to an insufficient biogenesis response in skeletal muscle”).

We apologize to the reviewer if he/she felt that the manuscript was not well written. Accordingly, it has been corrected by an English native speaker and we hope it will make our manuscript clearer. We also would like to thank the reviewer for his/her comments. We extensively revised our entire manuscript accordingly.

We conclude that “sepsis-induced muscle fiber atrophy and mitochondrial dysfunction were not explained by the energy debt that builds up at an early stage or the reduced physical activity”. The main objective of our study was to assess the implication of energy deficit on sepsis-induced mitochondrial dysfunction in skeletal muscle, which is known to be associated with prognosis in Humans [20]. While energy deficit is very common in the early phase of sepsis, its impact on mitochondrial function in skeletal muscle has never been studied. By mimicking the energy deficit observed during sepsis in a mouse model, we reported for the first time that sepsis-like energy deficit is not sufficient to induce mitochondrial dysfunction or muscle fiber atrophy. Besides this primary goal, our study, which was not designed to dissect a causal link between MtQc and mitochondrial dysfunction, brought additional data pointing to alterations of these pathways in skeletal muscle, in a cecal slurry model of sepsis. Other studies are required to better understand how these MtQC pathways are involved in mitochondrial dysfunction.

Of course they can not conclude with: Line 26-29 The authors write: ”energy deficit led to important metabolic adaptations – probably to better cope with this stress, not found in sepsis. Since hypocaloric nutrition is more and more used in Intensive Care Unit patients suffering from sepsis, concerns that energy deficit may worsen muscle status has to be considered in relation to our results”.

According to the reviewer’s comment, we changed this sentence as follows: ‘We demonstrated that a sepsis-like energy deficit was not in itself sufficient to alter muscle fiber atrophy or the mitochondrial population, in contrast with sepsis. On the other hand, energy deficit led to important metabolic adaptations which were not found in sepsis.’ Regarding the last sentence, instructions to the authors state that the lay summary should indicate ‘how the results will be valuable to society’. This is the reason why we believe that stating that ‘energy deficit may worsen muscle status has to be considered in relation to our results is important. We now discussed this point in the revised manuscript (l617-629).

The authors started from a clinical evaluation after cecal slurry injection. Line 255-256 the authors write that the temperature “slightly decreased in the SPF animals, compared to baseline”. If it is not statistically significant they can not write this sentence.

We thank the reviewer for his/her valuable comment. The temperature of SPF mice at H24 compared to baseline was statistically significant in a paired t-test. The manuscript has been modified accordingly.

Figure 1B. Present data as dots-box plot. The same should be done also for the panels D to G. Why the authors compare everything with 100% basal condition (0 hour)? Probably there is too much variability among mice at 0h. This variability could be a big problem.

We agree with the reviewer that variability could be a major issue. This is not the case in our study. We added a supplementary figure representing the clinical parameters observed at baseline in the form of Box and Whiskers plot (min – max) showing no significant difference between groups (New Figure S2).

Line 273-275: The authors write “Temperature and murine sepsis score (MSS) were higher in the sepsis group than the SPF group and were not different between SPF and SF groups except for the MSS at 24 hours which slightly increased in the SPF group” but the temperature in the sepsis group is LOWER.

We apologize for this mistake, which is now corrected.

Data in Figure 2 are not clear. The authors write:“Data are expressed as mean values with SEM (A, D) and SD (B, C, D, E) and analyzed by post-hoc Dunn’s test”. All the data should be presented as mean values with SEM. For results in D which error is shown: SEM or SD?

As suggested, we modified the expression of quantitative variables as mean values and SEM in order to clarify the reading of the figures.

Do the authors perform the Dunn’s test after a significant Kruskal-Wallis test?

As mentioned in the Methods (statistical analysis), data not following a normal distribution or with a small number were analyzed by a non-parametric Kruskal-Wallis test. If the latter was significant, then Dunn's post-hoc test was performed for multiple comparisons between the SPF vs. SF. and Sepsis vs. SPF groups. We have added test details in the legends of each figure.

More importantly analysis of covariance (ANCOVA) should be performed.

We believe that ANCOVA is not suitable for our study. ANCOVA is an analysis of the effect of independent categorical variables on a continuous dependent variable (by a combination of linear regression model and ANOVA) [21]. Instead, we used analysis of variance (ANOVA). When the data followed a normal distribution and had a sufficient number of patients, an ANOVA was used to determine whether the means of the SPF vs. SF and Sepsis vs. SPF groups were different. As the comparisons of means were done in pairs, a Fisher's LSD post-hoc test (which is a pairwise t-test with pooled variance) was then performed.

From Line 430: In the face of reduced food intake, SPF mice slightly decreased their energy expenditure similar to septic mice…… allowing us to study the intrinsic effect of sepsis or its energy deficit”. The authors based their study on a not significant data (Fig.2E). It is really difficult to understand.

We thank the reviewer for his/her comment, as this is a key point for understanding our experimental design. Our study is based on the assessment of the energy balance (energy expenditure minus energy intake, Figure 2.F) and not only on energy expenditure (Figure 2.E).

  • As mentioned in the methods (l138-139), a pair-feeding intervention was used to control the caloric intake of SPF mice. Each SPF mouse was matched with a Sepsis mouse of the same weight and was allowed to consume the same amount of food per 24 hours. Therefore, energy intake was identical between both groups (no significant difference, Figure 2.B).
  • The energy expenditure, measured by indirect calorimetry, was not experimentally controlled and did not differ between groups
  • The energy balance (expenditure minus intake) was then calculated. The energy balance of the SPF mice was decreased compared to SF mice (p<0.05), demonstrating an energy deficit. Finally, the energy balance was not statistically different between the Sepsis and SPF groups, which allows us to conclude that the energy deficit is the same in both groups.

To help the readers, we improved the section 2.2. Metabolic study.

In Fig3. The authors can not present the quantifications without showing representative images used for quantifications. There is not consistency in the way the data are showed. Why quantifications are performed in TA while this muscle is missing in the data of panels A and B?

As recommended by the reviewer, we have now included representative images (Figure 3.C). As previously mentioned, data are now expressed as means +/- SEM. Tibialis anterior muscle mass is displayed in the new Figure 3.A.

More importantly the authors conclude “sepsis-induced muscle atrophy cannot be explained by the sole energy deficit, indicating different underlying mechanisms” but they didn’t show the atrophy in muscle!

Figure 3.B,C provides readers with an accurate ex vivo assessment of muscle fiber atrophy in a mouse model. The evaluation of muscle fiber atrophy is based on the quantification of single muscle fiber size in a muscle cross-section [10,11]. Cross-sectional area (CSA) measurement, after immunofluorescence-based laminin labeling, is a robust method to measure muscle fiber size [12–15]. Thus, we analyzed a total of 148697 myofibers of the Tibialis anterior muscles with high-throughput acquisitions made by a slide scanner (Axioscan Z1, Zeiss, Germany) and the automatic and unbiased Muscle J macro (Mayeuf-Louchart et al. 2018). After the image quality check, Muscle J analyzed the CSA of each myofiber of an entire muscle section (>1000 myofibers per section and one triplicate per mouse). In our study, the TA CSA of Sepsis mice significantly decreased compared to the CSA of SPF mice. No differences were observed between SPF and SF mice. We also added representative images of the CSA immunostaining (Figure 3.C). In conclusion, the energy deficit was not sufficient to induce the muscle fiber atrophy observed during sepsis.

Accordingly, section 3.3 has been entirely rewritten.

As Buckinx et al. well described: there is no consensus on the best technique for measuring lean body mass [16]. We discuss the meaning of the lean mass and muscle wet weight in our model (l507-524). In particular, the muscles wet weights (l517-519), an indicator of muscle wasting [15], was not decreased in Sepsis mice compared to SPF or SF mice. This lack of change – in contrast to the CSA – could be explained by interstitial muscle swelling or muscle necrosis, which have been well-described in sepsis [17–19]. Such abnormalities may overestimate muscle weights and lean mass.

In Fig.4 the authors should show the OCR traces. What about the spare respiratory capacity ? Since Respiratory control rate changes independently of the substrate used this means that it is due to an intrinsic property of mitochondria. Please discuss this.

As suggested, we have now included representative JO2 flux traces from Datlab 7.4 software (Oroboros Instruments, Austria). Our experimental protocol was not designed to evaluate spare respiratory capacity. We rather preferred to explore the oxygen consumption rate depending on complex I and/or complex II, depending on glucose or fatty acid metabolism. In doing so, we provide the respiratory control ratio RCR, which is an indicator of mitochondrial OXPHOS efficacy. RCR was significantly reduced in Sepsis mice vs. SPF (Figure 4.B-D).

Since VDAC protein expression and mitochondrial DNA expression decreased in the Sepsis group compared with the SPF group (revised figure 5), the reduced mitochondrial biomass may partly explain the sepsis-induced mitochondrial dysfunction. In addition, the mitochondrial dysfunction – independent of the substrates and predominantly on complex I – may also be explained by a qualitative alteration of the OXPHOS proteins. Brealey et al. demonstrate that muscle nitrite/nitrate concentrations correlated with the complex I activity and the severity of the disease in septic shock patients [20]. Nitrosylation alters irreversibly complex I function [22,23] and reversibly the complex IV [24]. To support this hypothesis in our model, we showed that proteins harboring 3-nitrotyrosine are three-fold higher in the Sepsis group than the SPF group (new Figure S4). Others mechanisms implying mitochondrial quality control pathways that may also impair mitochondrial function are discussed in the revised manuscript.

Fig 5. The blots have poor quality. The OXPHOS are not clearly detected.

The bands are clearly detected and easily quantified with the software Image J. We have modified the contrast of the image for better visualization of the bands.

Line 376-7 the authors write “suggesting a preserved mitochondrial biomass in skeletal muscle of septic mice”. The authors must check in details whether or not mitochondrial mass is preserved in septic mice. Western blot detecting TOMM20 and CS should be informative.

We thank the reviewer for his/her valuable comment. We now added in our revised manuscript a mitochondrial biomass panel including VDAC protein expression (New Figure 5.A) and mitochondrial DNA level (New Figure 5.B) indicating that the mitochondrial biomass was reduced in the Sepsis group compared with the SPF group (revised Figure 5).

Line 379: “…tended to be lower in the sepsis group than the SFP group (p=0.08)”. The authors should increase the number of analyzed mice in order to understand if this tendency is corrector in not significant.

In the revised manuscript, we bring new data showing significant reductions in VDAC protein, mt DNA copy number, and Pgc1α mRNA in Sepsis compared with SPF mice. Overall, all these data are consistent and strengthen the fact that PGC1α signalling is likely reduced.

It is well known (for example check Mizushima and Yoshimori,Authophagy 2007; Kaludercic et al, Cardiovascular Research2020) that the comparison of LC3-I and LC3-II for ratio determinations may not be appropriate; rather, only the amount of LC3-II can be compared between samples.Then the authors should quantify only LC3II. Line 475-477 The authors conclude that “… the autophagy flux of septic mice was also deficient relative to SPF mice since LC3-II/I was lower and p62 higher – a marker of the accumulation of lysosomes-unfused autophagosomes”. The levels of autophagy or mitophagy cannot be assessed simply by studying the expression levels of the involved proteins and the blots of LC-II are in conclusive in this respect.  The findings could be corroborated with additional experiments, the most important of all would consist in a more direct demonstration that altered autophagy activity, and particularly of mitophagy, is affected. In particular Autophagy, lysosomal activity and mitophagy can be studied in a dynamic fashion in the presence and absence of specific inhibitors.

We agree with the reviewer’s comment that quantification of LC3 is not sufficient to assess autophagy. As referred to in Mizushima’s and Yoshimori's paper, LC3-II is an alternative way to evaluate the autophagy state. Accordingly, we added the analysis of LC3-II/GAPDH in the revised Figure 7.

As mentioned by the reviewer and described in the Guidelines for the use and interpretation of assays for monitoring autophagy (Klionsky et al. 2016, PMID 26799652), the use of autophagy inhibitors is the best way to assess autophagy flux. However, since our study was not designed to address this specific point and because most of the pharmacological autophagy inhibitors are not fully specific and may lead to animal death, further dedicated studies using genetically-modified mice should be designed to address this very interesting scientific question.

More importantly: Lines 90-92: The authors “aim at assessing if the negative energy balance commonly observed at the early phase of sepsis is involved in mitochondrial dysfunction and muscle wasting and whether it changes the MtQc”. The author should provide images of muscle sections and of mitochondria.

As asked by the reviewer, we added images of muscle sections in the revised Figure 3.C.

If the authors are not able to show mitochondrial morphology and at least they should analyze the levels of all the mitochondrial shaping proteins: Opa1 with long and short forms, phospho and total Drp1, Mfn1 and Mfn2 and Fis1. In particular Opa1 regulate complex V assembly and cristae function and morphology, independently of its fusion-related GTPase activity(patten et al., 2014). Moreover Opa1 is linked to autophagy.

We thank the reviewer for this very interesting point. Therefore, we designed new primers to study Mfn1, Mfn2, Opa1, Drp1 and Fis1. No changes were observed between groups except for Fis1. Fis1 mRNA increased by ~4-fold in the SPF compared with the SF group (p<0.05), and was lower in the Sepsis group than the SPF group (p<0.05) (Figure 6.A). We also performed additional western-blot: Mfn2 and Drp1 proteins (Figure 6.B) remained unchanged between groups. Thus, sepsis-like energy deficit mainly stimulated higher Fis1 expression, contrary to Sepsis, without modifying the other dynamics-related factors.

Moreover: Line 31 “Whether whole-body energy deficit participates in early skeletal muscle metabolism alteration has never been investigate”. Please check and discuss this recent paper: doi:10.1172/jci.insight.153944

We have carefully read the paper of Oh et al., which provides very exciting data. However, the authors do not discuss the impact of an energy deficit on skeletal muscle during sepsis but on hepatocytes. Therefore, it seems difficult to integrate this reference into our manuscript.

Reviewer 2 Report

The paper investigates early energy deficit events caused by sepsis in muscle mitochondria.

I like the idea of the research and the experimental design. Indeed, it is very important to know what causes energy deficit in septic subjects. The paper is logical and well written. I recommend minor revisions:

  1. It is unclear from the Abstract what animals were in each group; please reformulate this part.
  2. The headings of the Graphical abstract are confusing: "Energy homeostasis // Sepsis-associated energy deficit // Sepsis." I would expect the headings to contain uniform information.
  3. Figure 5 does not fit on the screen. The same problem exists for Table S1.
  4. Energy deficit is more pronounced at late stages of the sepsis. Please, provide a rationale for investigating early energy deficit events in your study.
  5. Please, provide information about reference gene(s) used for qPCR analysis. Primer sequences should be added to Table S3.
  6. There are a few minor grammatical errors, such as "All groups were resuscitated from the 12th hour to mimic human temporality." 

Author Response

Response to Reviewer 2

We thank this reviewer for his/her positive feedback, his/her important comments and appreciate the given opportunity to answer these concerns and thereby improve our manuscript.

The paper investigates early energy deficit events caused by sepsis in muscle mitochondria. I like the idea of the research and the experimental design. Indeed, it is very important to know what causes energy deficit in septic subjects. The paper is logical and well written. I recommend minor revisions:

  1. It is unclear from the Abstract what animals were in each group; please reformulate this part.

We thank the reviewer for his/her valuable comment and reworded the presentation of the groups in the Abstract (l22-24).

  1. The headings of the Graphical abstract are confusing: "Energy homeostasis // Sepsis-associated energy deficit // Sepsis." I would expect the headings to contain uniform information.

We thank the reviewer for his/her valuable comment and modified the graphical abstract for better clarity.

  1. Figure 5 does not fit on the screen. The same problem exists for Table S1.

We thank the reviewer for his/her attention and corrected these issues in the new manuscript.

  1. Energy deficit is more pronounced at late stages of the sepsis. Please, provide a rationale for investigating early energy deficit events in your study.

Within the first days of ICU admission, mitochondrial dysfunction is associated with poor prognosis in Humans [20]. Therefore, we aimed at assessing whether such mitochondrial dysfunction, here studied in skeletal muscle, could be related to the sole energy deficit. As shown in Figure S1, the first deaths occurred as early as the 36th hour in our sepsis model (l277). We focused just before this point to analyze both the mice that would survive and those that would die.

  1. Please, provide information about reference gene(s) used for qPCR analysis. Primer sequences should be added to Table S3.

We thank the reviewer for his/her valuable comment and added the information (reference gene) in the Table S3.

  1. There are a few minor grammatical errors, such as "All groups were resuscitated from the 12th hour to mimic human temporality."

According to the reviewer’s recommendation, our manuscript has been corrected by a native English speaker.

Reviewer 3 Report

The work from Pierre et al, adresses the effect of whole-body energy deficit in early skeletal muscle alteration, commonly associated during sepsis. The authors developed three animals models in which the control the caloric intake and induced sepsis to the mouse. The authors found that the caloric restriction did reproduce the early changes in the skeletal muscle induced by sepsis. Overall, the authors did a good work, very concise and with the precise experiments. However there are some missing points that could strengthen the conclusions

Major
The authors only addressed MFN2 and DRP1 as part of the mitochondrial dynamics. Why MFN1 and OPA1 were not evaluated? This could be an essential piece of information due to the role of MFN1 in the mitochondrial dynamics in the skeletal muscle and the role of OPA1 in the dynamics but also the cristae structure.

Interestingly, PGC1alpha is upregulated early in the SPF condition and there is a trend in the sepsis condition; however, no changes were found in the mitochondrial mass. It would be useful as well to monitor the mtDNA content.

Since most of the changes are very early, I would recommend to perform qPCR for the most relevant genes (Mito dynamics, PGC1a, Mito mass markers), this because changes in the transcriptome most of the times are faster than the changes in the proteome

Minor
-In the introduction, only MFN2 and DRP1 are mentioned as part of mitochondrial dynamics. Why MFN1, OPA1, and DRP1 anchors were omitted? Please update

-Also, it is interesting how the sepsis condition induces changes in mitochondrial respiration without causing changes in the overall complexes mass. The authors should provide a brief explanation in the discussion about the possible mechanism

Author Response

Response to Reviewer 3

The work from Pierre et al, adresses the effect of whole-body energy deficit in early skeletal muscle alteration, commonly associated during sepsis. The authors developed three animals models in which the control the caloric intake and induced sepsis to the mouse. The authors found that the caloric restriction did reproduce the early changes in the skeletal muscle induced by sepsis. Overall, the authors did a good work, very concise and with the precise experiments. However there are some missing points that could strengthen the conclusions.

We thank this reviewer for his/her positive feedback, his/her important comments and appreciate the given opportunity to answer these concerns and thereby improve our manuscript.

Major
The authors only addressed MFN2 and DRP1 as part of the mitochondrial dynamics. Why MFN1 and OPA1 were not evaluated? This could be an essential piece of information due to the role of MFN1 in the mitochondrial dynamics in the skeletal muscle and the role of OPA1 in the dynamics but also the cristae structure.

We thank the reviewer for his/her valuable comment. Therefore, we explored mitochondrial dynamics: Mfn1, Mfn2, Opa1, and Drp1 mRNAs (Figure 6.A), as well as Mfn2 and Drp1 proteins (Figure 6.B) remained unchanged between groups. Nevertheless, Fis1 mRNA increased by ~4-fold in the SPF compared with the SF group (p<0.05), and was lower in the Sepsis group than the SPF group (p<0.05) (Figure 6.A). Thus, sepsis-like energy deficit mainly stimulated higher Fis1 expression, contrary to Sepsis.

Interestingly, PGC1alpha is upregulated early in the SPF condition and there is a trend in the sepsis condition; however, no changes were found in the mitochondrial mass. It would be useful as well to monitor the mtDNA content.

We fully agree with the reviewer’s comment. We performed additional analyses of VDAC protein expression and the mitochondrial DNA content by the ratio Nd1/Ppia and Nd2/Ppia to better monitor mitochondrial biomass. VDAC1/3 (revised Figure 5.A) did not differ between the SF group and SPF group, but were lower in the Sepsis group than in the SPF group (p<0.05). In addition, the expression of Mt-Nd1 and Mt-Nd2 normalized to the nuclear Ppia were similar in the SF and SPF groups but lower in the Sepsis group than the SPF group (p<0.05), indicating that sepsis induced mt DNA depletion (revised Figure 5.B). Thus, the sepsis-like energy deficit is not responsible for the reduced mitochondrial biomass observed in Sepsis. Regarding mitochondrial biogenesis, Pgc1α mRNA expression decreased in the Sepsis group compared with the SPF group (p<0.05) (revised Figure 5.C). mRNA expressions of Nuclear Respiratory Factor 1 (Nrf1), Mitochondrial transcription factor A (Tfam), and Silent mating type information regulation 2 homolog 1 (Sirt1) – regulators of biogenesis – remained unchanged between groups (SPF vs. SF and Sepsis vs. SPF). Total Pgc1α protein expression increased 3.9-fold in the SPF compared with the SF group (p<0.01) and remained unchanged in the Sepsis group compared with SPF mice (revised Figure 5.A). Taken together, these results indicate that, within 24 hours, sepsis starts decreasing the biogenesis signaling and mitochondrial biomass and induced an impaired Pgc1α response relative to the negative energy balance

Since most of the changes are very early, I would recommend to perform qPCR for the most relevant genes (Mito dynamics, PGC1a, Mito mass markers), this because changes in the transcriptome most of the times are faster than the changes in the proteome.

We agree that changes in the transcriptome are faster than in the proteome. Consequently, we performed additional experiments to evaluate the relative mRNA expression of the biogenesis- (described above), dynamics- and autophagy-related actors. We will find in the revised manuscript the following information. “Mfn1, Mfn2, Opa1, and Drp1 mRNAs (New Figure 6.A), as well as Mfn2 and Drp1 proteins (Figure 6.B) remained unchanged between groups. Nevertheless, Fis1 mRNA increased by ~4-fold in the SPF compared with the SF group (p<0.05), and was lower in the Sepsis group than the SPF group (p<0.05) (New Figure 6.A). Thus, sepsis-like energy deficit mainly stimulated higher Fis1 expression, contrary to Sepsis.

The mRNA expression of Bnip3 increased 3.4-fold in the SPF group compared with the SF group (p<0.05), and was lower in the Sepsis group than the SPF group (p<0.05) (New Figure 7.A). Although Beclin1, Atg7, and Atg12 in the Sepsis group were all slightly higher than the SFP group (p<0.05), Unc-51-like autophagy activating kinase (Ulk1), Atg5, Atg8l, and Lamp2 were not different (Figure 7.A). The activated form of Parkin (phospho-Ser65 Parkin) and Pink1 were not different between groups (SPF vs. SF or Sepsis vs. SPF) (Fig 7.B). LC3B-II was not different between groups (SPF vs. SF and Sepsis vs. SPF). Relative protein ratios of LC3B-II : LC3B-I and protein expression of p62 (a marker of the accumulation of autophagosomes not fused with lysosomes) were not different between SPF and SF groups but decreased by 50% and increased threefold, respectively, in the Sepsis group compared with SPF (p<0.05). (Fig 7.B). Overall, different mitophagy-related responses were observed between SPF and Sepsis groups”

We also discuss the link between Fis1 and the mito-autophagy process according to our new results.

Minor
-In the introduction, only MFN2 and DRP1 are mentioned as part of mitochondrial dynamics. Why MFN1, OPA1, and DRP1 anchors were omitted? Please update

As suggested, we updated the introduction mentioning Fis1 as Drp1 anchors, Mfn1, and Opa1 for a better understanding of the manuscript.

-Also, it is interesting how the sepsis condition induces changes in mitochondrial respiration without causing changes in the overall complexes mass. The authors should provide a brief explanation in the discussion about the possible mechanism.

We thank the reviewer for his/her relevant comment and fully agree. We now revised and improved our discussion accordingly. Please now read lines 565-616 from the revised discussion. Briefly, we discussed about post-translational modifications and how the different mitochondrial quality control pathways may explain this functional alteration without a major reduction in mitochondrial proteins.

In particular, nitrosylation of OXPHOS proteins may occur during sepsis since Brealey et al. demonstrate that muscle nitrite/nitrate concentrations correlated with the complex I activity and the severity of the disease in septic shock patients [20]. Nitrosylation alters irreversibly complex I function [22,23] and reversibly complex IV [24]. To support this hypothesis in our model, we showed that proteins harboring 3-nitrotyrosine are three-fold higher in the Sepsis group than the SPF group (New Figure S4).

Round 2

Reviewer 1 Report

I'm sorry but I stand by my idea. As presented, the work opens up many questions instead of presenting conclusions supported by experimental evidence.

I think that the authors cannot use “sepsis-like energy deficit”, since it is more correct to speak about energy deficit (same food restriction as for septic mice: it is a “fasting” protocol, then the crucial point is: how much food the mice ate?)

The authors have maintained this sentence: ”Since hypocaloric nutrition is increasingly used in Intensive Care Unit patients suffering from sepsis, concerns that a sepsis-like energy deficit may worsen muscle status should be considered in relation to our findings”. It is a conclusion that is not supported by the data and it must be deleted. The food reduction/fasting did not affect muscles!

 Clearly the authors have modified the images in Fig.3c. The figure in fact exhibits some type of suspicious image alteration and it cannot be acceptable since it is potentially fraudulent. Notably, the cross sectional comparison among the three groups is crucial to conclude that “Muscle fiber atrophy was not explained by the sepsis-like energy defect”. Even in the title is stated that ”Sepsis-like energy deficit is not sufficient to induce early muscle fiber atrophy…”

The authors discuss:” High-levels of Fis1 promote mitochondrial fragmentation and trigger autophagy. Taken together,  our data suggest that sepsis impaired the MtQC leading to an insufficient renewal of the mitochondrial population in skeletal muscles”. This is a very interesting point and the authors should focus on autophagy processes. Then they should really demonstrate if autophagy flux is reduced upon sepsis and induced in the SPF group. Moreover the data about LC3 are not clear. According the presented data in Fig.7 LC3II is not significantly different among the groups. As LC3I is not detectable in many samples showed in the panel in Fig.7b(from the left samples 4-6-7-9- 14 and 15)  how was possible for the authors to calculate the ratio?

It is difficult to interpret the mitochondrial biogenesis data. The authors tell that” an increase in Pgc1α-dependent pathway” was observed in the SPF group. Why Mfn2 a well known target of pgc1a was not found upregulated?(Fig.6).

In conclusion the three major processes (atrophy, mitochondrial biogenesis and autophagy/mitophagy) are not well clarified and relative findings are not supported by experimental data.

Reviewer 3 Report

The manuscript from Pierre et al. shows significant improvement. Pleased find below my major and minor comments.

Major

In figure 3, one of the panels looks out of focus.

In the same figure, the conclusion about skeletal muscle fiber atrophy is a little bit confusing since the authors didn’t find any changes in the wet weight of the different muscles. This needs to be clarified. Also, it would enrich the manuscript more details on skeletal muscle atrophy, like the measurement of the muscle atrophy-related genes or the determination of the type of fiber that is more susceptible to the atrophy induced by acute sepsis. 

Minor comments

In the introduction, the authors should be more specific in describing the concepts of an early or long phase.

In the introduction, the authors introduce Fis1 as THE adaptor of DRP1; however, since 2014, the adaptors Mid49/Mid51 and Mff were discovered and demonstrated the relevance for mitochondrial fission, showing that it could be even more specific for mitochondrial fission that fis1, this Fis1 is also involved in peroxisome fission. The literature should be updated with the field's most recent papers/reviews.

In the methods, the authors should provide more details about the diet used and how the caloric restriction was performed.

In the results, the authors should be more specific and clarify the criteria to determine sepsis's early and long phases. Maybe a table could be helpful with the checkpoint to determine the early or long phase. 

In the results, How the authors explain the change in the body mass for the SPF, without showing changes in the muscle mass. Is this only attributed to changes in the fat mass?

In the results, I suggest including a table in the supplementary data with no normalized data so then is possible to see the actual changes in the body mass parameters of the animals.
